# Migration of Alpine Slavs and machine learning: Space-time pattern mining of an archaeological data set

Benjamin Štular[1]☉*, Edisa Lozić[1,2]☉, Mateja Belak[1‡], Jernej Rihter[1‡], Iris Koch[2‡], Zvezdana Modrijan[1‡], Andrej Magdič[3‡], Stephan Karl[2‡], Manfred Lehner[2‡], Christoph Gutjahr[2‡]

**1** Znanstvenoraziskovalni Center Slovenske Akademije Znanosti in Umetnosti, Ljubljana, Slovenia, **2** Institute of Classics, University of Graz, Graz, Austria, **3** Območna Enota Maribor, Javni Zavod Republike Slovenije za Varstvo Kulturne Dediščine, Maribor, Slovenia

☉ These authors contributed equally to this work.
‡ MB, JR, IK, ZM, AM, SK, ML and CG also contributed equally to this work.
\* benjamin.stular@zrc-sazu.si

**Citation:** Štular B, Lozić E, Belak M, Rihter J, Koch I, Modrijan Z, et al. (2022) Migration of Alpine Slavs and machine learning: Space-time pattern mining of an archaeological data set. PLoS ONE 17(9): e0274687. https://doi.org/10.1371/journal.pone.0274687

**Data Availability Statement:** All relevant files are available from the Zenodo database at https://zenodo.org/record/5813527 (doi: 10.5281/zenodo.

## Abstract

The rapid expansion of the Slavic speakers in the second half of the first millennium CE remains a controversial topic in archaeology, and academic passions on the issue have long run high. Currently, there are three main hypotheses for this expansion. The aim of this paper was to test the so-called "hybrid hypothesis," which states that the movement of people, cultural diffusion and language diffusion all occurred simultaneously. For this purpose, we examined an archaeological Deep Data set with a machine learning method termed time series clustering and with emerging hot spot analysis. The latter required two archaeology-specific modifications: The archaeological trend map and the multiscale emerging hot spot analysis. As a result, we were able to detect two migrations in the Eastern Alps between c. 500 and c. 700 CE. Based on the convergence of evidence from archaeology, linguistics, and population genetics, we have identified the migrants as Alpine Slavs, i.e., people who spoke Slavic and shared specific common ancestry.

## 1. Introduction

On Easter Monday, 2 April 568 CE, Alboin, son of Audoin and king of the Lombards, led his people on an arduous journey. They were to "abandon the barren fields. . . and come and take possession of Italy." Because from the "west and north [Italy] is shut in by the range of Alps, they reached it from the eastern side by which it is joined to Pannonia" (Paul the Deacon, Book II, Ch. V. and IX.; [1]).

Historiography records Alboin's success: He founded the Kingdom of Italy the next year and his successors ruled there for almost two centuries, e.g., [2]. However, historiography is less clear about events in the lands that Alboin and his people left behind. These lands include the Eastern Alps; we know little more than that this region was sparsely populated by Romans

5813527) and https://zenodo.org/record/5761811 (doi: 10.5281/ZENODO.5761811).

**Funding:** The sources of funding that have supported the work are Austrian Science Fund grant number I 3992 (Initials of authors who received the grant: M.L., E.L., I.K., C.G., S.K.) and Javna Agencija za Raziskovalno Dejavnost RS grant number J6-9450 (Initials of authors who received the grant: B.Š., M.B., Z.M., J.R., A.M.). The funders had no role in study design, data collection and analysis, decision to publish, or preparation of the manuscript.

**Competing interests:** The authors have declared that no competing interests exist.

in the sixth century and that several "peoples" or *gentes*, e.g. Ostrogoths, Gepids, and Slavs, attempted to make it their home, e.g., [3, 4]. Of these, the Slavs were the most successful.

The rapid spread of the Slavic language in the second half of the first millennium CE remains a controversial topic. There are two main reasons for this. First, the lack of first-hand, written sources before the end of the ninth century. Unlike the Lombards, the Slavs had no Paul the Deacon to recount their early history. Second, archaeological evidence on this subject is sparse compared to many other Early Medieval "peoples". As a result, there is a "propensity for sweeping explanations" [5].

Currently, there are three main hypotheses for the spread of Slavic between about 400 and 850 CE, e.g., [6, 7]. The first hypothesis assumes that speakers moved in all directions from their small original habitat, the so-called *Urheimat*, e.g., [8–10]. The second hypothesis assumes the diffusion of the Slavic cultural model among non-Slavic populations or, in its extreme form, the diffusion of language alone, e.g., [7, 11–14]. Many archaeologists adhere to the third, hybrid hypothesis. The hybrid hypothesis states that movement, cultural diffusion, and language diffusion occurred simultaneously [15–18]. This is supported by recent research in population genetics and linguistics. It seems that the language spread in the West Slavic zone mainly by migration to sparsely populated areas, and in the East Slavic zone by a combination of migration and language shift. The spread in the South Slavic region was triggered by migration, but the main mechanism for further spread was a language shift from local Balkan idioms to Slavic [19].

We adhere to the hybrid hypothesis in its most recent form [19], which is based primarily on population genetics and language studies. The aim of this paper was to test this hypothesis with archaeological data from the Eastern Alps. The specific objective of the paper was to elucidate the settlement of the Alpine Slavs, as the Slavic-speaking Early Mediaeval population of the Eastern Alps is known in historiography [4, 20–22]. To this end, we applied the technique called "space-time pattern mining" to examine a large archaeological data set from the period 400 to 1100 CE. In doing so, we have developed two archaeology-specific methodological innovations that can be applied to archaeological studies of any period.

## 2. Material and methods

### 2.1. Data set and study area

Our data set is Zbiva [23], which is an open access research database for the archaeology of the Eastern Alps in the Early Middle Ages (in our study 600 to 1100 CE). The inception of the database in the early 1980s was deeply rooted in the scientific research context of that time, which determined both the geographical and chronological focus of the data set [24].

Currently, Zbiva contains data on 3,379 sites and 11,596 related bibliographic units in more than half a million database fields. Because the data set is the result of four decades of deliberate scholarly work and attentive curation, it is best described by the concept of Deep Data. The Deep Data approach is one in which we make full use of all the information available in the data to gain the knowledge [25], *cf.* [26]. As far as the authors are aware, Zbiva is unmatched in Slavic archaeology, and the only comparable data set for Early Medieval archaeology is Open-Atlas [27], with its affiliate, THANADOS [28].

Recently, the team behind Zbiva has enriched the database, adding new information with a focus on chronology and location. The chronology of each site was re-evaluated by an expert, using modern typochronologies based on C14 dates [29–31]. The locations were also improved using maps (historical and modern) and satellite imagery available through open access web GIS applications. In addition, the data set was enriched with metadata (e.g., the

confidence level for chronology and location) and paradata (e.g., sources for dating). Furthermore, it was expanded to include Late Antiquity sites (400 to 600 CE in our study).

However, this data enrichment focused on a geographically limited subset of data. This subset of 1,105 archaeological sites constitutes the study area of this article. It includes present-day Slovenia, southern Austria (Carinthia, Styria, East Tyrol, parts of Salzburg and Upper Austria) and a small part of northern Italy (the Trieste region) (Fig 1).

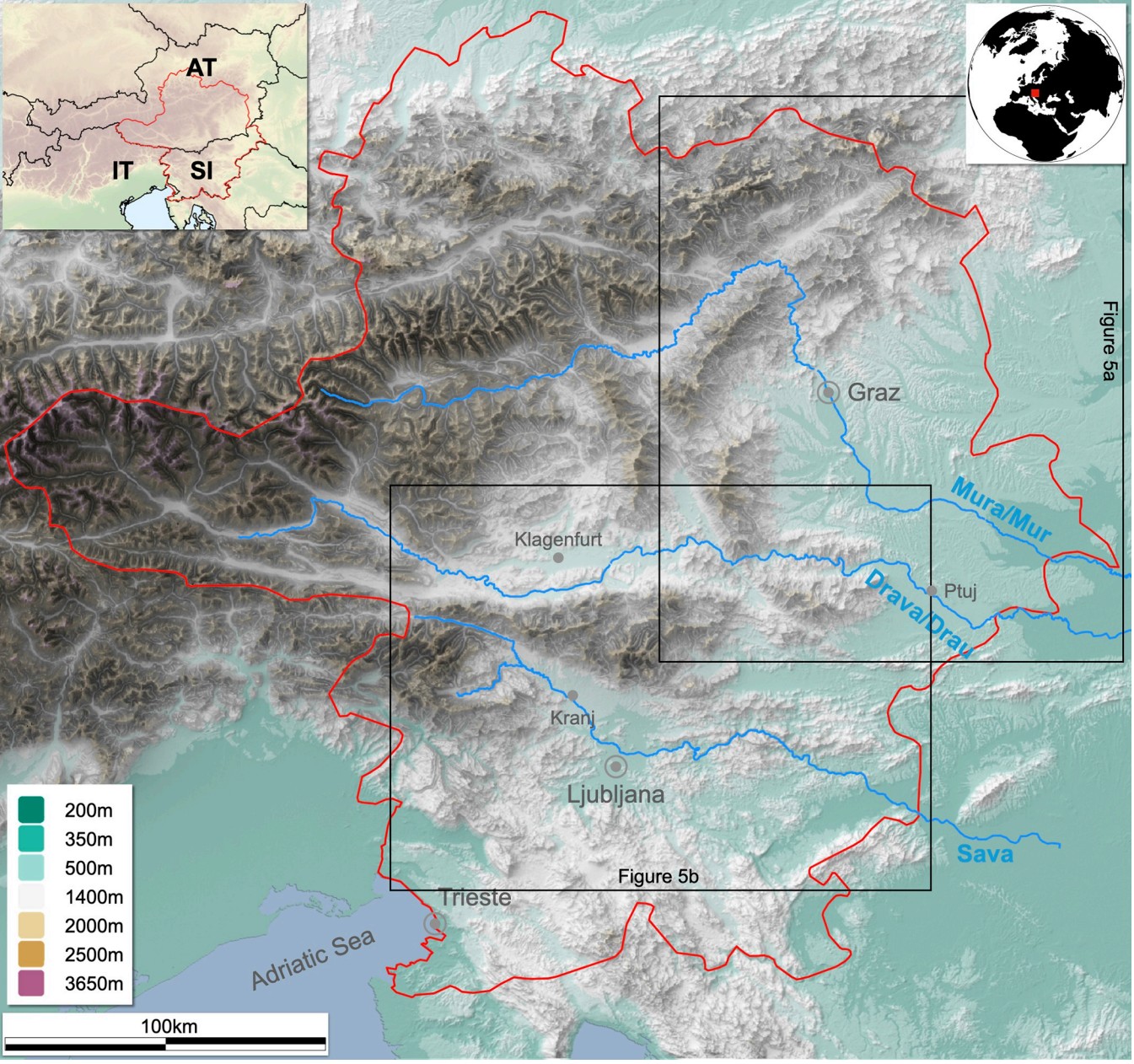

**Fig 1.** Map of the study area (upper left corner Lat. 48.22015, Lon. 12.35667; lower right corner Lat. 45.29785, Lon. 16.41784). The study area is marked in red and locations of Fig 5A and 5B in black (authors E.L. and B.Š; contains information from OpenStreetMap and OpenStreetMap Foundation, which is made available under the Open Database License; contains information adapted and modified from Copernicus Land Monitoring Service product EU-DEM25, which was produced with funding by the European Union).

The data are described in more detail in the data paper [25] and are openly accessible via the Zenodo repository [32].

Compared to the entirety of Slavic territories ours is a small study region. But this region is an excellent (if not pivotal) case study for understanding the general processes of the spread of Slavic speakers in Early Middle Ages for three reasons. (i) Archaeologically, this is the only region where data is readily available for advanced spatial analysis, including machine learning (see above). (ii) The historiographical sources are second to none and include the oldest permanent Slavic political entity (Carantania, after 650 CE; e.g., [4, 22, 33–35]), the oldest Slavic text other than the canonical Old Church Slavic (ninth-century *Monumenta Frisingensia*; [36]), and the oldest mention of a member of a specifically Slavic social elite, a *župan* (iopan Physso, 777 CE [21, 22, 37]). (iii) Linguistically, the area is on the southwestern periphery of the spread of Slavic, bordering Germanic and Romance languages; this is important because peripheries typically preserve archaisms better than centres.

## 2.2. Space-time cube

Artificial Intelligence (AI) is becoming an ever more integral part of the digital humanities. Here, we focus on machine learning, which is a subset of AI. It refers to a set of data-driven algorithms and techniques that automate data prediction, classification, and clustering. Machine learning is rapidly being adopted by archaeologists interested in the analysis of geospatial data, material culture, texts, and artistic data [38, 39] and is most often used in combination with Big Data. Among the most prolific fields for machine learning within archaeology is airborne LiDAR data, e.g. overview in [40].

Currently, most machine learning processing takes place in the Phyton and R programming environments. This means that it requires coding and is therefore not readily accessible to most archaeologists. In this paper, we have used the only off-the-shelf pipeline that can be readily applied to archaeology: The toolset for space-time pattern mining in ArcGIS Pro v. 2.9 (ESRI, Redlands CA, USA). A similar pipeline has already been demonstrated on archaeological data [41], but to our knowledge ours is the first time this method was used to test a specific archaeological hypothesis.

Some comments on terminology are in order. "Space Time Pattern Mining" is a commercial name for the toolset used in this work [42]. We classify our method as machine learning, following [39]. They list the methods used in the cultural heritage domain, which are similar to the time series clustering, under unsupervised machine learning. In addition, machine learning is used in the background of this software to enable, for example, intelligent data-driven defaults [43].

Why use space-time pattern mining? Whenever archaeologists (or anyone, for that matter) look at a map, we inherently begin to turn that map into information by finding patterns and assessing trends. However, sometimes the patterns in the data are too complex to be clustered, observed and aggregated by the human eye. In such cases, we can use space-time pattern mining tools to answer our questions confidently, objectively, and repeatably.

The first step in space-time pattern mining is to organize the data. In this case, the data is incorporated into what is known as the "space-time cube model." This model is based on Torsten Hägerstand's time geography, which introduces the time axis into the traditional Cartesian coordinate system, e.g., [44]. It can be thought of as a three-dimensional cube consisting of space-time fields, where the dimensions x and y represent space and the dimension z represents time (Fig 2A). The data is stored in the so-called "space-time NetCDF cube." The process of constructing the space-time cube data model is complex and beyond the scope of this article, but it is well documented, e.g., [41].

The concept of the space-time cube is familiar to archaeologists. At its core, it is an application of the way we perceive (or have perceived in the past) archaeological excavations: In

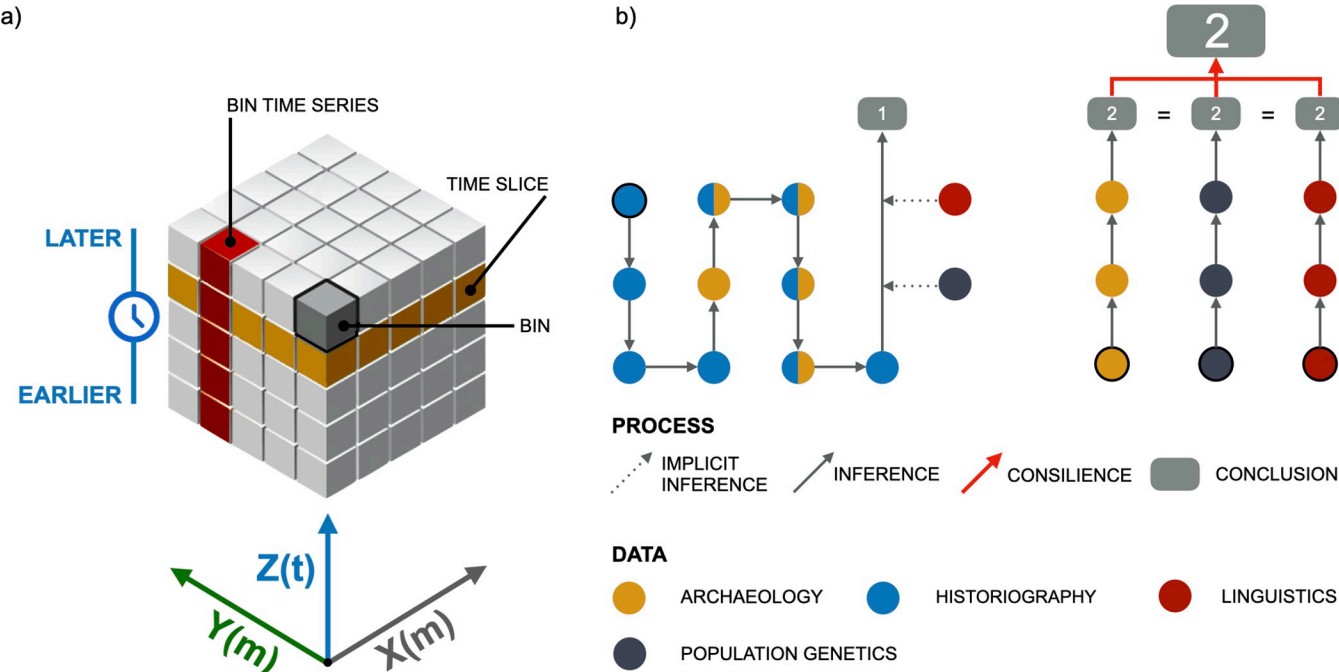

**Fig 2.** Graphical representations of selected methods: a) space-time cube model (after ESRI); b) comparison of two types of inference: On the left, data from different fields are compiled to draw a unified conclusion (analysis of Pohl's [17] interpretation of our study region as an example), and on the right, consilience.

spatial quadrants and time phases. The quadrants in the xy grid are constant throughout the excavation, and the phases stack on top of each other, with the earliest at the bottom and the latest at the top.

For our case study, we aggregated data into 5 km big (y dimension) and 25 years long (z dimension) hexagonal bins. The size of the hexagon was chosen as the largest in which the relevant processes can be observed; in Early Medieval archaeology, it roughly corresponds to the site catchment area of a single settlement, e.g., [45]. The time interval of 25 years was determined on the basis of the data properties. The time sensitivity of relevant archaeological dating is about half a century, i.e., ± 25 years. However, start and end dates can often be a quarter of a century, rarely as brief as decades or even years. The accuracy of our dates is therefore 50 years, but the precision is approximately 25 years. Accordingly, 25-year intervals were chosen for analysis, but the accuracy of the data requires that archaeological interpretation be limited to 50-year intervals.

This method is very sensitive to the difference between no data (areas not analyzed) and null data (areas analyzed, but not found to contain any known sites). To account for this, we limited the space-time cube to the area used for data collection. Additionally, we excluded areas higher than 1,400 m above sea level (Fig 1: shades of brown). In our region, this altitude delineates the highest valley settlements from the lowest high-mountain pastures. The latter were excluded from the analysis because they are specialized seasonal settlements that were always dependent on valley settlements.

## 2.3. Time series clustering

Clustering is one of the most widely used machine learning techniques in the field of cultural heritage [39]. Its goal is to organize similar data into homogeneous groups or clusters. Clusters

are formed by grouping objects that have maximum similarity with other objects within the cluster and minimum similarity with objects in other clusters. For large and complex data sets, unsupervised approaches offer the best solution. Time series clustering is a type of unsupervised clustering used for data with a temporal component [46, 47].

The concept of time- series clustering is deeply familiar to archaeologists having been used since the nineteenth century. At that time, for example, the three-age system, which divides the development of human civilization into the Stone Age, Bronze Age, and Iron Age, was defined by clustering similarly dated stone/bronze/iron artefacts. As most archaeologists know from experience, such clustering is relatively easy for a few artefacts or sites but becomes daunting when the numbers run into the hundreds or thousands of objects or sites. In such cases, unsupervised time series clustering can be used.

In this article, we applied time series clustering to classify sites into chronological groups. In each group, the chronology (start date, end date) of the sites is more similar to each other than that of the sites outside the group.

The similarity between the clusters is measured by the so-called "pseudo-F statistic." The larger the pseudo-F value, the more different each cluster is from the other clusters [48]. There are several ways to calculate the pseudo-F statistic, each depending on which characteristics of the time series are considered important. In our experience, the most appropriate for archaeology is the "Profile (Fourier)" method, i.e., method based on Fourier series periodic function. It is used to cluster time series that have similar, smooth, and periodic patterns over time [42].

This method lends itself to the analysis of archaeological processes because they usually follow a consistent pattern: A gradually introduced innovation is followed by a peak of use and a steady decline. Thus, archaeological processes can be compared to seasons, where temperature follows a consistent annual pattern, with higher temperatures in summer and lower temperatures in winter. The "Profile (Fourier)" method is best suited to finding locations that have the most similar annual temperature patterns, for example, to distinguish between locations with mild and severe winters. A season in this example represents an archaeological phase or period.

In our case, we opted to ignore the range, i.e., the magnitude of the values in each period. To extend the analogy above, ignoring the range causes the change of seasons in two places occurring at the same times to be considered similar, even though the actual temperatures are different.

## 2.4. Modified emerging hot spot analysis

Spatial analysis is often called upon to determine the density of observed phenomena, and one of the most common tools to do this in archaeology is the so-called "hot spot analysis." It uses the Getis-Ord $G^*$ statistic to calculate z-scores and p-values within a given spatial neighborhood. These indicate whether the observed spatial clustering of high and low values is more (hot spot) or less (cold spot) pronounced than would be expected from a random distribution, e.g., [49].

In this article, we have used an emerging hot spot analysis that examines the clustering of high and low values over time, in addition to spatial trends. The space-time cube is evaluated bin-by-bin, and each bin is analyzed relative to its space-time neighbors. Thus, each site is related not only spatially but also temporally to neighboring sites. The result is similar to the traditional hot spot analysis, except that it is in 3D (where the z dimension represents time).

Such a result can deliver an overwhelming amount of information. Therefore, the tool evaluates the trends of hot spots and cold spots over time using the Mann-Kendall trend test, e.g., [50] and categorizes each location in the study area accordingly. For example, a location is

considered a consecutive hot spot if it has an uninterrupted series of statistically significant hot spot bins over the latest time step intervals, but less than 90% of the total [42]. The resulting 2D representation of trends can be termed a "trend map."

However, the trend map provided by the tool is not suitable for archaeology for two reasons. First, it assumes that the latest records are the focus of analysis. Second, it was designed for data sets much larger than ours and those of most archaeological studies.

We therefore modified the trend map by focusing on chronological periods previously calculated by the time series clustering method. For example, a location was considered a first period consecutive hot spot if it had an uninterrupted hot spot series of at least 100 years within the first period (detailed description in S1 Table). We term this an "archaeological trend map".

In emerging hot spot analysis, the spatial and temporal neighborhoods have a significant influence on the results. We found, through empirical observation, that the best results for hot spots and cold spots were obtained with different settings. For cold spots: fixed distance method with 20 km neighborhood, and time step three. For hot spots: k-nearest neighbors (kNN) method with six spatial neighbors, time step one [42].

Therefore, we have introduced another archaeology-specific modification to the tool. For the purposes of this article, we superimposed the cold spots derived using the first settings with the hot spots derived using the second settings in a single visualization. We refer to this method as "multiscale emerging hot spot analysis."

To ensure the highest level of methodological transparency, reproducibility, and transparency and to reduce the time researchers spend replicating the work of other research groups, we provide the ready-to-re-use data in GIS format and the GIS protocol (S1 Appendix).

## 3. Theory

### 3.1. Consilience

The specifics of archaeological inference, e.g., [51], are not often outlined in articles such as ours. In this case, however, it is necessary because academic passions on the question of the migration of the Slavs have long been running high, and the methods of inference are often scrutinized. Moreover, this topic is invariably interdisciplinary, but the interdisciplinarity is achieved through a variety of approaches.

In order to enrich this discussion with the most objective archaeological information possible, we have chosen to base our inference on consilience of induction. Consilience, also known as "convergence of evidence," is a scientific principle that states that the same conclusion is much stronger when drawn from independent and unrelated sources. Confidence is strongest when evidence from different fields is considered because the methods and/or data are different [52].

Although it is rarely referred to by its name, this principle is popular in archaeology. For example, consilience is applied whenever radiocarbon dating is invoked to support archaeological dating.

It is important to distinguish between consilience, where conclusions are drawn independently before being correlated, and the more common interdisciplinary approach to the study of Slavic migrations, where data from different fields is compiled to draw a unified conclusion with a mix and match approach (Fig 2Bb). To this end, we have been careful to consider only information from each field that has not been influenced by findings from another field. For example, in the Discussion we consider linguistic information [53], but disregard the conclusions drawn on the same subject matter using supporting evidence from archaeology [54]. We

also take care to include only interpretations reached by domain specialists, as reinterpretations by non-specialists can be problematic [6].

## 3.2. Material culture as ethnicity, identity, and habitus?

The main archaeological argument for the migrations of the Slavs is based on the association of the Slavs with various archaeological cultures or *habitus*. For instance, the archaeological assemblages of the so-called Prague Culture are associated with the Early Slavs, e.g., [55–61].

From a modern theoretical perspective, this argument draws on Pierre Bourdieu's [62] notion of *habitus*; its basic premise is that practical knowledge is embodied in daily practices and that material culture, including pottery, expresses these practices, e.g., [63]. However, equating a *habitus* (e.g., the archaeological assemblages of Prague culture) with a people/tribe/ethnicity (e.g., the Early Slavs) is an additional step that Bourdieu did not anticipate. In much of the literature after the mid-1960s, the notion that material culture is more or less directly related to cognition of peoples was questioned by many; the acceptance that archaeological cultures simply cannot be directly correlated with ethnicity took hold, e.g., [64, 65].

Rather than engage in this discourse, we based our argument only on the categories of archaeological data that are indisputable: Location and chronology of the site. Our inference thus eclipses theoretical issues about the associations of material culture with ethnicity, e.g., [66], identity, e.g., [67], or habitus, e.g., [63]. Whether or not the Prague Culture archaeological assemblages are associated with the Early Slavs was immaterial to our conclusions.

## 4. Results

### 4.1. Archaeological periodization

The result of our time series clustering is the archaeological periodization of sites. The most archaeologically meaningful result is when the data is clustered into three periods (pseudo-F statistic value 421.595). These are the Late Antiquity and two Early Middle Ages periods (Fig 3). The latter two correspond to the traditional periodization of the jewelry into the Carantanian and the Köttlach phases, e.g., [68] or groups A/B and C of Eichert [69].

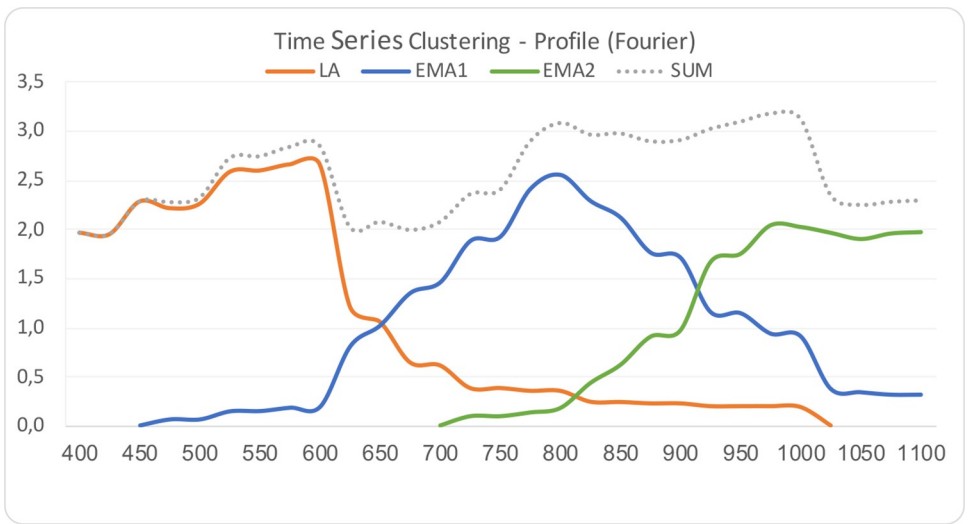

**Fig 3. Archaeological periodization with time series clustering: LA—Period 1, Late Antiquity; EMA1—Period 2, Early Middle Ages 1; EMA2—Period 3, Early Middle Ages 2.** Values on x axis are years CE, values on y axis are unitless and relative.

Three important conclusions may be drawn from these results. The first is, that the general increase in sites between 400 and 500 CE and the decrease after 1000 CE does not reflect reality, as we know it from numerous sources, e.g., [22, 70]. Rather, it reveals the weakness of the underlying data set: The period before 500 CE has not been collated systematically, and data for the period after 1000 CE (which, until recently, was not considered relevant to archaeology in the region, e.g., [71]) is lacking. Regardless, the present data set is suitable for the study of the half millennium between 500 and 1000 CE, which was the aim.

Second, unlike changes in material culture, changes in landscape are more gradual and often overlap. For example, the time series of Late Antiquity does not end until 1000 CE, as some sites exhibit continuity from Late Antiquity onward (for example, the town of Kranj, e.g., [72]). The results of time series analysis in archaeology are therefore complex and must be interpreted with great care.

Third, we substantiated the long-established periodization of the Early Middle Ages into two periods by an independent source of data: The chronology of sites rather than the typology of jewelry. This, then, is the first quantitative evidence that changes in jewelry styles taking place in the second half of the 9th century were reflected in changes in the archaeological landscape. The most likely explanation is that both changes had the same underlying cause, which however is beyond the scope of this article.

## 4.2. Archaeological landscape

The emerging hotspot analysis revealed an astonishing quantity and quality of information (Fig 4). Most relevant to our topic are the extensive areas of cold spots in the northern part of the region and the general patchiness, i.e. activity is concentrated in enclaves. In this respect, the archaeological landscape between 500 and 1000 CE differs from both the preceding Roman period settlement and subsequent High Medieval period, which both exhibit a more regular pattern of settlement. This is important in providing a context for understanding various historical processes. For example, the reason it is so difficult for historiographers to define the exact borders of *Carniola*, e.g., [3] and *Carantania*, e.g., [4] is that in the patchy landscape precise fixed borders most likely never existed.

The main focus of this article was migration. The two tools for detecting migrations in our data were provided by Curta [13]. First, migration must have occurred if settlements and cemeteries suddenly appear in a previously sparsely populated area, i.e. cold spots are immediately followed by hot spots. Second, migration can also be detected by the sudden appearance of a material culture without local traditions or parallels in a given area.

With the first tool, migration was documented in the easternmost part of the study area. In the period between 450 and 500 CE this is a cold spot area, but after c. 500 CE hot spots appear along the river Mura (*Ger.* Mur). After a period of consolidation until c. 600 CE, a series of small-scale neighbourhood migrations upstream of the Mura and the adjacent Drava (*Ger.* Drau) rivers is documented by numerous hot spots (Fig 5A; S2 Appendix).

With the second tool, a migration upstream of the Sava river after c. 675 CE was documented. Between c. 600 and 675 CE, there was a gradual decline in hot spots along the Sava, but between c. 675 and 750 CE there was a reversal of that trend (Fig 5B; S2 Appendix). This trend reversal alone could be explained by other causes than migration. The evidence for migration was provided by the time series clustering, which showed a sudden and complete shift in material culture: the number of Late Antiquity sites diminishes dramatically and at the same time Early Medieval sites start appearing (Fig 3). This shift has long been known in archaeology as the transition from fortified hilltop settlements to unfortified lowland settlements, e.g., [70], which determines the transition from Late Antiquity to the Early Middle Ages.

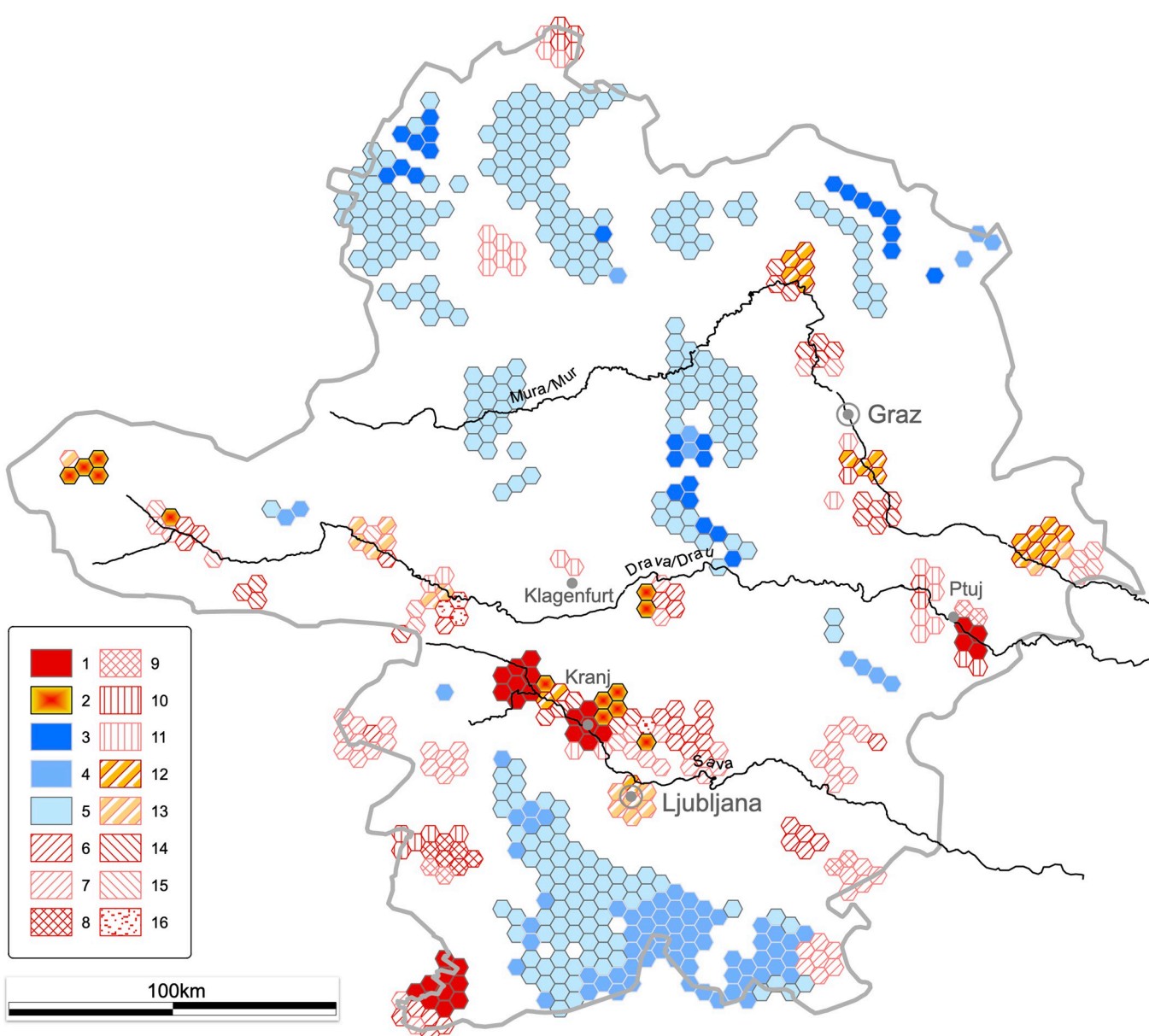

**Fig 4. Archaeological trend map of the modified categorization of the multiscale emerging hot spot analysis.** See S1 Table for the legend (authors E.L. and B.Š; contains information from OpenStreetMap and OpenStreetMap Foundation, which is made available under the Open Database License; contains information adapted and modified from Copernicus Land Monitoring Service product EU-DEM25, which was produced with funding by the European Union).

On the basis of this data, a comment can be made on the size of the migrations. Overall, hot spots interpreted as resulting from the first migration account for only 4% of all hot spots in c. 500 CE, which indicates a relatively small founder population. However, by c. 700 CE, 59% of hot spots can be interpreted as resulting directly or indirectly from both migrations. Although this is a very rough estimate, far from giving a direct indication of the actual number of people involved, it is the best available and by far the most tangible to date, *cf.*, [3, 4, 37, 73–76]. As such, it is an invaluable foundation for explaining acculturation processes following migrations, which, however, are not the subject of this article.

a)

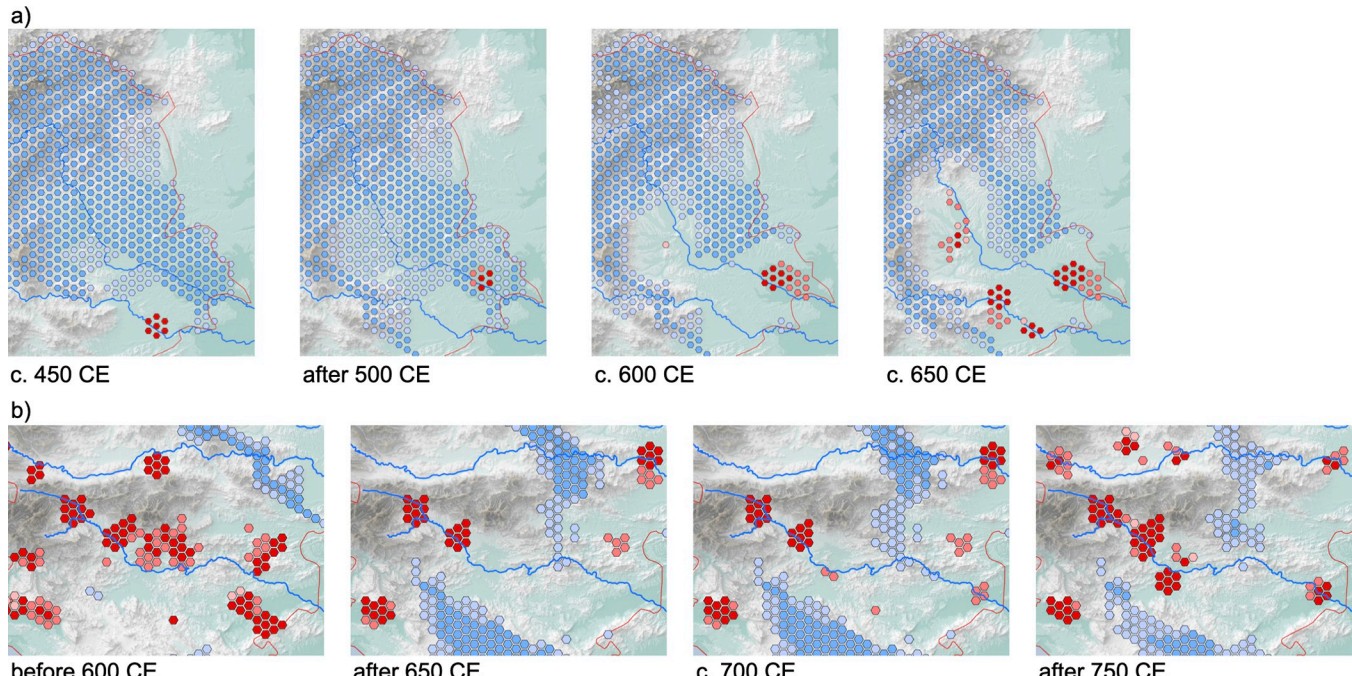

**Fig 5.** Time slices from the emerging hot spot analysis for selected areas: a) Eastern part of the study area from 450 CE to 650 CE; b) central part of the study area from before 600 CE to after 750 CE (authors E.L. and B.Š; open access raw data sources used: EU-DEM v1.1, https://land.copernicus.eu; OpenStreetMap, https://www.openstreetmap.org).

## 5. Discussion

### 5.1. Archaeology

Our data thus testifies to two migrations: The first upstream of the Mura and Drava rivers after c. 500 CE, and the second before c. 700 CE upstream of the Sava river. This is an important discovery, but it does not shed light on who the migrants were. Based on the archaeological [77–80]) and historiographical [17, 20, 22] context, we can hypothesize that they were Early Slavs. But this hypothesis inherits all the weaknesses of the existing ones, which are based on the controversial presence of archaeological assemblages of Prague Culture and scant written sources.

The hypothesis may, however, be considered the null hypothesis that can be tested with the consilience principle. The new archaeological evidence for two separate migrations allows us to correlate it with interpretations from the linguistics and genetics of modern populations. These two fields of science use completely different data sources and methods than archaeology and have recently made significant advances in understanding Slavic migrations.

### 5.2. Linguistics

Let us first take a look at linguistics. While a language or dialect may be tied to any number of identities within a given period, a shared linguistic innovation requires a linguistic community, for which the term "founder population" has been proposed [6, 54].

Modern Slovenian, which is spoken today in the southern part of the research area, belongs to the South Slavic clade, according to the traditional classification of the Slavic languages [81]. There are, however, considerable linguistic similarities between the Slovenian and the West

Slavic lects. These similarities were explained either by the existence of a specific link between Slovenian and West Slavic or by a mixed South and West Slavic origin of Slovenian [82–86].

Specific ties between Slovenian and South Slavic on the one hand and West Slavic on the other have recently been demonstrated with a series of phylogenetic NeighborNet networks. The analysis concluded that Slovenian seems to be almost equally close to the West and South Slavic, but distant from the East Slavic, "thus supporting the putative mixed nature of Modern Slovenian" [87]. It is the latter interpretation that interests us.

In conclusion of the above cited analysis further studies of Slovenian dialects are proposed in order to clarify the position of Slovenian among the Slavic languages. One of such study examined the diatopic distribution and semantic development of *glčěti as the primary neutral verb meaning 'to speak'. It was carried from an emergent dialect of Slavic and is now wide-spread in present-day central Russia, central Bulgaria, and in Slovenia along the Mura and Drava rivers. Of interest is the hypothesis of possible relationships between the "early Slavic speakers who spoke dialects in which *glčěti played a central role as a verb of speech" and those who did not within modern Slovenia, i.e. the southern part of our study area. The hypothesis states that this dichotomy, together with the -ny- ‖ -nǫ- isogloss, "can be viewed as inherited pre-migration cleavages" [53], that is, "the dialects of Slavic brought to the subalpine area. . . differed (amongst themselves)" ([6]; translated from the Slovenian by B.Š.; the subalpine area mentioned corresponds approximately to our study region). Since "shared linguistic innovation presupposes a community" ([6]; translated from the Slovenian by B.Š.), it follows that heterogeneous dialects presuppose heterogeneous communities or founder populations.

Therefore, the linguistic interpretations imply that the southern part of the region under study was originally populated by two founder populations that spoke two heterogeneous Slavic dialects. One, using *glčěti, populated areas along the rivers Mura and Drava, the other one populated areas further west.

## 5.3. Genetic history

Second, let us turn to genetic history. This scientific field attempts to reconstruct human evolution and the history of human populations using genetic information obtained from either modern or ancient DNA [88]. DNA has been described as a document containing "messages from the past" [89] and is a proven tool in prehistoric archaeology, e.g., [90–93]. However, there are significant obstacles to the use of modern DNA when it comes to Late Antiquity and the Middle Ages. For example, the historical population-level information that this method reflects is complex and overlapping and should not be understood as representing a direct correspondence between population history and social history [94, 95]. For the study of this time period, ancient DNA or ancient genomic DNA data are more appropriate, e.g., [96, 97]. However, ancient DNA data are not available in sufficient quantity to study the expansion of Slavic speakers.

Regardless of the methodological shortcomings, it is agreed that the results of genetic studies on modern DNA are indisputable in terms of providing information on genetic proximity and can contribute to hypotheses about human population history, including Late Antiquity and Early Mediaeval migrations [94, 97–99].

The most complete study of modern DNA pertaining to the expansion of Slavic speakers examined all ethnic groups living today who speak Balto-Slavic languages, as well as their neighbors. It concluded that the genetic diversity of today's Slavs was predominantly formed *in situ* (i.e., the substrate genetic components in the settled areas prevail), with marked differences between West and East Slavs on the one hand and South Slavs on the other. However, there is genetic affinity showing a common ancestry (i.e., a homogeneous genetic substrate

inherited from the ancestral population) among the Slavs, which probably demonstrates the historical dispersion of a once uniform population [87, 100]. This was recently confirmed in a review article, which concluded, that the migration of Slavs was accompanied by active assimilation of indigenous European populations [101].

Looking more closely at the region under study, the variability of the microsatellite loci of the Y chromosome is telling. It shows that the inhabitants of present-day Slovenia are far removed from all other South Slavic populations [87]. When this was first discovered in an earlier study it was interpreted to "suggests that at least two different migration waves of the Slavs may have reached the Balkans in the early Middle Ages" [102].

Considering only the modern Slovenian population, there is one possible ancestral haplogroup for all Slovenian populations which has the highest frequency along the Mura and Drava rivers. This could indicate that the origin of this ancestral haplogroup was in this area and that it later spread westward [103].

Thus, population genetic studies show that the southern part of the study region was possibly settled by two separate migrations. The earlier one took place along the Mura and Drava rivers; the later one was to the west of that area.

### 5.4. Consilience

Interpretations from three scientific fields, using completely different data sets and methods, shows a consilience or convergence of evidence. Archaeological, linguistic, and genetic evidence suggests, with varying degrees of certainty, that there were two separate migrations to the southern part of the region under study: The earlier one along the Mura and Drava rivers, and the later one, which archaeology can locate along the Sava river. Archaeology and genetics suggest that acculturation was the predominant post-migration process. Linguistics and genetics indicate that the migrants were Slavs. In particular, linguistics indicates that the migrants were speakers of Slavic, and genetics confirms that they had a homogeneous genetic substrate inherited from a single ancestral population common only to ethnic groups speaking Slavic today.

Based on consilience, we can define the immigrants who arrived in the Eastern Alps between c. 500 and c. 700 CE with by far the highest reliability to date. They were speakers of Slavic and shared a specific "Slavic" ancestry. Our archaeological analysis places these migrations in space and time with some precision (Fig 5).

## 6. Conclusions

The aim of this article was to test the hybrid hypothesis of Slavic migration using archaeological data, with the Alpine Slavs as a case study. The term "migration" in the title was deliberately chosen to be somewhat provocative, as modern historiography and archaeology of the Early Middle Ages tend to downplay the role of physical migrations.

We used selected machine learning methods to analyze an archaeological data set that can be described as Deep Data. Specifically, we used two methods: Time series clustering and a modified emerging hot spot analysis. The former method is directly suitable for archaeology without modification, whereas the latter required two archaeology-specific modifications: The archaeological trend map and the multiscale emerging hot spot analysis.

The results have provided us with an overwhelming quality and quantity of new information. In this article, we have focused on confirming two separate migrations of the Alpine Slavs. Based on the convergence of evidence from archaeology, linguistics, and population genetics, we define the immigrants as Alpine Slavs who were speakers of Slavic and shared

specific "Slavic" ancestry. Two founder populations migrated to the Eastern Alps: The first after c. 500 and the second before c. 700 CE.

The identities and ethnicity of the migrants (as defined by modern historiography) are, in our view, beyond the scope of archaeology. The acculturation processes that took place after the migration will be discussed elsewhere. From the available evidence, however, it is clear that the crucial process was cultural spread *sensu* Heather [18]. We envisage that the number of migrating people was relatively small and more akin to a small group infiltration than a mass migration. The movement itself was a part of it, but the processes that took place afterwards were historically the most important.

Thus, we have achieved the aim of the article, which was to prove the validity of the hybrid hypothesis of Slavic migration with archaeological data. The migration of the Alpine Slavs was a combination of movement of people, cultural diffusion, and language diffusion, all occurring simultaneously. This clearly refutes the hypothesis that only the cultural model or even only the language spread.

While this paper focused on a specific question related to the migration of Slavs, the methods we developed, borrowing and adapting from a variety of disciplines, can be applied to archaeological studies of any period, anywhere that suitable data is available. We hope that these advances will be used beneficially by other scholars and establish a new, practical approach to add to the archaeological arsenal of methodologies.

In the field of machine learning in archaeology and the digital humanities in general, we hope to have shown that, in addition to Big Data, Deep Data also holds great potential.

## Supporting information

**S1 Table. Archaeological trend map, detailed description of 16 classes.**
(DOCX)

**S1 Appendix. GIS Protocol for multy-scale emerging hot spot analysis.**
(DOCX)

**S2 Appendix. Animation of the emerging hot spot analysis time slices from 400 CE to 1100 CE (authors E.L. and B.Š; contains information from OpenStreetMap and OpenStreetMap Foundation, which is made available under the Open Database License; contains information adapted and modified from Copernicus Land Monitoring Service product EU-DEM25, which was produced with funding by the European Union).**
(GIF)

## Acknowledgments

The authors give thanks for the constructive comments and suggestions by academic editors and the anonymous reviewers.

## Author Contributions

**Conceptualization:** Benjamin Štular, Edisa Lozić.

**Data curation:** Mateja Belak.

**Formal analysis:** Edisa Lozić.

**Funding acquisition:** Benjamin Štular.

**Investigation:** Mateja Belak, Jernej Rihter, Iris Koch, Zvezdana Modrijan, Andrej Magdič, Stephan Karl.

**Methodology:** Benjamin Štular, Edisa Lozić.

**Project administration:** Benjamin Štular, Andrej Magdič, Manfred Lehner, Christoph Gutjahr.

**Software:** Benjamin Štular.

**Supervision:** Benjamin Štular, Manfred Lehner.

**Validation:** Edisa Lozić.

**Visualization:** Edisa Lozić.

**Writing – original draft:** Benjamin Štular, Edisa Lozić.

**Writing – review & editing:** Benjamin Štular, Edisa Lozić, Andrej Magdič, Stephan Karl, Manfred Lehner.

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
