## [Decision Letter · Decision Letter 0]

24 May 2022

PONE-D-22-00071Migration of Alpine Slavs and Machine Learning: Space-Time Pattern Mining of an Early Medieval Data Set from the Eastern AlpsPLOS ONE

Dear Dr. Štular,

Thank you for submitting your manuscript to PLOS ONE. After careful consideration, we feel that it has merit but does not fully meet PLOS ONE’s publication criteria as it currently stands. Therefore, we invite you to submit a revised version of the manuscript that addresses the points raised during the review process. Both Reviewer 1 and 2 have a number of minor comments that should help to improve the exposition. Reviewer 2 additionally have more general comments that would minimally require you to present a more balanced assessment of the strength of the evidence.

We look forward to receiving your revised manuscript.

Kind regards,

Søren Wichmann, PhD

Academic Editor

PLOS ONE

Journal Requirements:

“This research was funded by Austrian Science Fund (FWF) grant number I 3992, and Slovenian Research Agency (ARRS) grant numbers J6-9450 and P6-0064.”

4. We note that Figures 1, 5 & S3 in your submission contain [map/satellite] images which may be copyrighted. All PLOS content is published under the Creative Commons Attribution License (CC BY 4.0), which means that the manuscript, images, and Supporting Information files will be freely available online, and any third party is permitted to access, download, copy, distribute, and use these materials in any way, even commercially, with proper attribution. For these reasons, we cannot publish previously copyrighted maps or satellite images created using proprietary data, such as Google software (Google Maps, Street View, and Earth). For more information, see our copyright guidelines: http://journals.plos.org/plosone/s/licenses-and-copyright.

a. You may seek permission from the original copyright holder of Figures 1, 5 & S3 to publish the content specifically under the CC BY 4.0 license. 

Reviewers' comments:

Reviewer's Responses to Questions

**Comments to the Author**

1. Is the manuscript technically sound, and do the data support the conclusions?

Reviewer #1: Yes

Reviewer #2: Partly

2. Has the statistical analysis been performed appropriately and rigorously? 

Reviewer #1: Yes

Reviewer #2: I Don't Know

3. Have the authors made all data underlying the findings in their manuscript fully available?

Reviewer #1: Yes

Reviewer #2: Yes

4. Is the manuscript presented in an intelligible fashion and written in standard English?

Reviewer #1: Yes

Reviewer #2: Yes

5. Review Comments to the Author

Reviewer #1: 111-112: "Spread of Common Slavic" is a little awkward - "Common Slavic" usually refers to the more or less uniform construct established by the comparative method, but ancillary methods such as etymology and dialect geography have demonstrated that "Late Common Slavic" was dialectally heterogeneous. Perhaps say "Spread of Slavic" and denote a time period.

112: Perhaps consider a 4th reason: center-periphery phenomena. Peripheries typically preserve archaism better than centers. Slovene is on the southwestern periphery of the spread of Slavic.

114: so strong links > such strong links

117: Old Church Slavic (capitalize, as it is used as a proper name). Better would be "canonical Old Church Slavic" (= corpus of Cyrillo-Methodian tradition texts), which excludes the Monumenta Frisingensia (as you intend).

118: a gloss and/or explanation of župan would help the uninitiated reader.

429-430: Consider also:

Kushniarevich, Alena and Kassian, Alexei, “Genetics and Slavic Languages”, in: Encyclopedia of Slavic Languages and Linguistics Online, Editor-in-Chief Marc L. Greenberg. Consulted online on 11 April 2022 <http: 10.1163="" 2589-6229_eslo_com_032367="" dx.doi.org="">

First published online: 2020

432: "West Croats" is not clear: coastal (= Čakavian)? NW (= Kajkavian)? Note also that "White Croats" (Slovak: Bieli Chorváti) are identified in the medieval West Slavic region. Nobody has established who the "White Croats" were -- only the name is known and there is no evidence specifically linking them with (or denying linkage with) the South Slavic Croats.</http:>

Reviewer #2: General evaluation

The study examines the spread of Slavic in the Eastern Alps around 500-1000 CE. It argues that archaeological, linguistic and genetic evidence suggests that there were two migrations into this area between 500 and 700 CE, and that the migrants spoke Common Slavic. As a historical linguist with only a rudimentary knowledge of archaeology and population genetics, I will limit myself to discussing the linguistic aspects of the study under review.

I find the topic of the study very interesting. However, I am not convinced that the linguistic evidence referred to by the authors is strong enough to support the conclusions they draw.

Accordingly, I would encourage the authors to either modify their confidence in their analysis of the linguistic evidence or to leave out entirely the linguistic component of the paper. I am aware that especially the latter option will make the paper less attractive as the linguistic perspective is important for the overall interest of the study.

An alternative, and in my opinion much more satisfying, solution would be to strengthen the linguistic dimension of the study. Instead of the vague and sporadic references to secondary literature (often in Slovenian, which – sadly – makes it accessible to only a minority of the readers of the journal), it would be interesting to see actual analyses of the linguistic evidence that the authors claim to have for their conclusions. This would probably require that a person with a thorough knowledge of the linguistic background be included in the group of authors, if such a person is not already present.

In the attached pdf file containing the study I have added my comments on specific parts. The file also includes corrections and stylistic suggestions.

6. PLOS authors have the option to publish the peer review history of their article (what does this mean?). If published, this will include your full peer review and any attached files.

Reviewer #1: No

Reviewer #2: No

---

## [Author Response · Author response to Decision Letter 0]

14 Jul 2022

PONE-D-22-00071

Migration of Alpine Slavs and machine learning: space-time pattern mining of an archaeological data set

PLOS ONE 

Rebuttal letter

Dear Editor, dear reviewers

The authors would like to thank the reviewers for their thorough and thoughtful reviews. We have implemented all the suggested minor comments that helped us to improve the exposition of the article. We have also addressed reviewer 2's general comments by introducing a more balanced assessment of the strength of the linguistic evidence.

Below are our responses, point by point (reviewers remarks in grey, our responses in black).

Reviewer #1:

111-112: "Spread of Common Slavic" is a little awkward - "Common Slavic" usually refers to the more or less uniform construct established by the comparative method, but ancillary methods such as etymology and dialect geography have demonstrated that "Late Common Slavic" was dialectally heterogeneous. Perhaps say "Spread of Slavic" and denote a time period.

Corrected:

... the spread of Slavic in Early Middle Ages…

112: Perhaps consider a 4th reason: center-periphery phenomena. Peripheries typically preserve archaism better than centers. Slovene is on the southwestern periphery of the spread of Slavic.

Corrected:

114: so strong links > such strong links

Corrected.

117: Old Church Slavic (capitalize, as it is used as a proper name). Better would be "canonical Old Church Slavic" (= corpus of Cyrillo-Methodian tradition texts), which excludes the Monumenta Frisingensia (as you intend).

Corrected:

...canonical Old Church Slavic...

118: a gloss and/or explanation of župan would help the uninitiated reader.

Corrected:

... the oldest mention of a member of a specifically Slavic social elite, župan,

429-430: Consider also:

Kushniarevich, Alena and Kassian, Alexei, “Genetics and Slavic Languages”, in: Encyclopedia of Slavic Languages and Linguistics Online, Editor-in-Chief Marc L. Greenberg. Consulted online on 11 April 2022 

First published online: 2020

Corrected:

The reference has been added.

432: "West Croats" is not clear: coastal (= Čakavian)? NW (= Kajkavian)? Note also that "White Croats" (Slovak: Bieli Chorváti) are identified in the medieval West Slavic region. Nobody has established who the "White Croats" were -- only the name is known and there is no evidence specifically linking them with (or denying linkage with) the South Slavic Croats.

Corrected:

It shows that the inhabitants of present-day Slovenia (and partly those from western Croatia) are far removed from all other South Slavic populations.

Reviewer #2 

I find the topic of the study very interesting. However, I am not convinced that the linguistic evidence referred to by the authors is strong enough to support the conclusions they draw.

Accordingly, I would encourage the authors to either modify their confidence in their analysis of the linguistic evidence or to leave out entirely the linguistic component of the paper. I am aware that especially the latter option will make the paper less attractive as the linguistic perspective is important for the overall interest of the study.

An alternative, and in my opinion much more satisfying, solution would be to strengthen the linguistic dimension of the study. Instead of the vague and sporadic references to secondary literature (often in Slovenian, which – sadly – makes it accessible to only a minority of the readers of the journal), it would be interesting to see actual analyses of the linguistic evidence that the authors claim to have for their conclusions. This would probably require that a person with a thorough knowledge of the linguistic background be included in the group of authors, if such a person is not already present.

The authors have decided to modify our confidence in the interpretations of the linguistic evidence to which we refer in our work.

In addition, we have:

- expanded and substantiated our presentation of the cited linguistic interpretations;

- in the interests of a balanced article, we have also expanded and substantiated our presentation of the genetic history evidence;

- key interpretations from linguistics and genetics have been replaced by direct quotations to make it clearer that we are only using existing interpretations by domain specialists.

In the attached pdf file containing the study I have added my comments on specific parts. The file also includes corrections and stylistic suggestions.

We thank the reviewer for corrections and stylistic suggestions; we have implemented them in full, so there are now stylistic changes throughout the text.

Below are rebutals to the inline comments, listed by lines.

l.59: Corrected.

l.85: Corrected.

l.107: Corrected.

l.110-119: The entire paragraph has been rewritten to address the reviewer's comments. Especially with regard to the linguistic evidence, we have followed the recommendation of reviewer 1. It now reads:

“Compared to the entirety of Slavic territories ours is a small study region. But this region is an excellent (if not pivotal) case study for understanding the general processes of the spread of Slavic speakers in Early Middle Ages for three reasons. (i) Archaeologically, this is the only region where data is readily available for advanced spatial analysis, including machine learning (see above). (ii) The historiographical sources are second to none and include the oldest permanent Slavic political entity (Carantania, after 650 CE; e.g., [4,22,34-36]), the oldest Slavic text other than the canonical Old Church Slavic (ninth-century Monumenta Frisingensia; [37]), and the oldest mention of a member of a specifically Slavic social elite, a župan (iopan Physso, 777 CE [21,22,38]). (iii) Linguistically, the area is on the southwestern periphery of the spread of Slavic, bordering Germanic and Romance languages; this is important because peripheries typically preserve archaisms better than centres.«

l.156: Corrected.

l.262: Corrected, it now reads: »Moreover, this topic is invariably interdisciplinary, but this interdisciplinarity is achieved through a variety of approaches.«.

l.276: Corrected.

l.282: Corrected.

l.320: Corrected.

l.390-406: The entire subsection 5.2 has been rewritten and expanded to accomodate the reviewers comments. It now reads:

“Let us first take a look at linguistics. Linguistics holds that, while a language or dialect may be tied to any number of identities within a given period, a common linguistic innovation requires the community in which it occurs, the so-called founder population [6].

Modern Slovenian, which is spoken today in the southern part of the research area, belongs to the South Slavic Clade, according to the traditional classification of the Slavic languages [82]. There are, however, considerable linguistic similarities between the Slovenian and the West Slavic lects. These similarities were explained either by the existence of a specific link between Slovenian and West Slavic or by the mixed South and West Slavic origin of Slovenian [83-87].

Specific ties between Slovenian and South Slavic on the one hand and West Slavic on the other have recently been demonstrated with a series of phylogenetic NeighborNet networks. The analysis concluded that Slovenian seems to be almost equally close to the West and South Slavic, but distant from the East Slavic, "thus supporting the putative mixed nature of Modern Slovenian" [33]. It is the latter interpretation that interests us.

In conclusion of the above cited analysis further studies of Slovenian dialects are proposed in order to clarify the position of Slovenian among the Slavic languages. One of such study examined the diatopic distribution and semantic development of *gъlčěti as the primary neutral verb meaning 'to speak'. It was carried from an emergent dialect of Proto-Slavic and is now widespread in present-day central Russia, central Bulgaria, and in Slovenia along the Mura and Drava rivers. Of interest is the hypothesis of possible relationships between the "early Slavic speakers who spoke dialects in which *gъlčěti played a central role as a verb of speech" and those who did not within modern Slovenia, i.e. southern part of our study area. The hypothesis states that this dichotomy, together with the -ny- || -nǫ- isogloss, "can be viewed as inherited pre-migration cleavage" [54], that is, "the dialects of Slavic brought to the subalpine area ... differed (amongst themselves)" ([6]; translated from the Slovenian by B.Š.; the subalpine area mentioned corresponds approximately to our study region). Since "shared linguistic innovation presupposes a community" ([6]; translated from the Slovenian by B.Š.), it follows that heterogeneous dialects presuppose heterogeneous communities or founder populations.

Therefore, the linguistic interpretations imply that the southern part of the region under study was originally populated by two founder populations that spoke two heterogeneous Slavic dialects. One, using *gъlčěti, populated areas along the rivers Mura and Drava, the other one populated areas further west.”

l.441: Corrected.

l.446: Corrected, it now reads: »Archaeology, linguistics, and population genetics have each deduced with varying degrees of certainty that there were…«.

l.450-1: Corrected

l.450-2: The statement in the text is: »...linguistics confirms that the migrants were speakers of Common Slavic, ...« Reviewers comment is: »this is an overstatement«. Our statement is based on the text in 5.2 (now expanded) with references. To our knowledge, there are no scientific publications stating the contrary. 

We therefore believe to have met the reviewers criteria by expanding the section 5.2.

l.455: See comment above.

l.458: See comment above.

l.460: Corrected.

l.477: Corrected.

---

## [Decision Letter · Decision Letter 1]

30 Aug 2022

PONE-D-22-00071R1Migration of Alpine Slavs and machine learning: space-time pattern mining of an archaeological data setPLOS ONE

Dear Dr. Štular,

Thank you for submitting your manuscript to PLOS ONE. The reviewer has a few comments, mainly on style issues. The comments are in the PDF (see pp. 43 ff. in the PDF). Please respond to those. If your revision is careful it may not be necessary to send out the paper for vetting again.

We look forward to receiving your revised manuscript.

Kind regards,

Søren Wichmann, PhD

Academic Editor

PLOS ONE

Journal Requirements:

Reviewers' comments:

Reviewer's Responses to Questions

**Comments to the Author**

1. If the authors have adequately addressed your comments raised in a previous round of review and you feel that this manuscript is now acceptable for publication, you may indicate that here to bypass the “Comments to the Author” section, enter your conflict of interest statement in the “Confidential to Editor” section, and submit your "Accept" recommendation.

Reviewer #2: All comments have been addressed

2. Is the manuscript technically sound, and do the data support the conclusions?

Reviewer #2: Yes

3. Has the statistical analysis been performed appropriately and rigorously? 

Reviewer #2: I Don't Know

4. Have the authors made all data underlying the findings in their manuscript fully available?

Reviewer #2: Yes

5. Is the manuscript presented in an intelligible fashion and written in standard English?

Reviewer #2: Yes

6. Review Comments to the Author

Reviewer #2: The revised version addresses most of my concerns regarding the previous version. In the attached PDF file I have added a few comments, mostly regarding stylistics.

In a few instances (indicated in the PDF file) I recommend the authors to modify their formulations. I do not insist, however – my recommendations are meant as a help to the authors to avoid criticism for sounding too confident in their conclusions regarding the linguistic evidence (which I think could be interpreted in alternative ways).

7. PLOS authors have the option to publish the peer review history of their article (what does this mean?). If published, this will include your full peer review and any attached files.

Reviewer #2: No

---

## [Author Response · Author response to Decision Letter 1]

31 Aug 2022

PONE-D-22-00071

Migration of Alpine Slavs and machine learning: space-time pattern mining of an archaeological data set

PLOS ONE 

Rebuttal letter

Dear Editor, dear reviewers

The authors would like to thank the reviewer #2 for their thorough and thoughtful reviews. We have implemented all the suggested comments in full.

Below are our revisions.

line 402-404

Let us first take a look at linguistics. Linguistics holds that, while a language or dialect may be tied to any number of identities within a given period, a common linguistic innovation requires the community in which it occurs, the so-called founder population [6].

Let us first take a look at linguistics. While a language or dialect may be tied to any number of identities within a given period, a shared linguistic innovation requires a linguistic community, for which the term "founder population" has been proposed [6, 55].

Comment: the term "founder population" was proposed by Greenberg and the text has been changed to reflect this; also, the reference has been amended accordingly.

line 406

...the South Slavic Clade,...

...the South Slavic clade,...

line 409

... by the mixed South and West Slavic origin...

... by a mixed South and West Slavic origin...

line 422

... i.e. southern part...

... i.e. the southern part...

line 424

... pre-migration cleavage"...

... pre-migration cleavages"...

line 475-476

Archaeology, linguistics, and population genetics have each deduced with varying degrees of certainty that there were two separate migrations...

Archaeological, linguistic, and genetic evidence suggests, with varying degrees of certainty, that there were two separate migrations…

line 480-481

Linguistics and genetics have interpreted that the migrants were Slavs. In particular, linguistics confirms that the migrants were speakers of Slavic, and genetics confirms that they had a...

Linguistics and genetics indicate that the migrants were Slavs. In particular, linguistics indicates that the migrants were speakers of Slavic, and genetics confirms that they had a…

line 485

Table 1 has been removed.

line 606

Updated reference of an article that is now in press.

---

## [Editor Report · Decision Letter 2]

2 Sep 2022

Migration of Alpine Slavs and machine learning: space-time pattern mining of an archaeological data set

PONE-D-22-00071R2

Dear Dr. Štular,

We’re pleased to inform you that your manuscript has been judged scientifically suitable for publication and will be formally accepted for publication once it meets all outstanding technical requirements.

Kind regards,

Søren Wichmann, PhD

Academic Editor

PLOS ONE
---

## [Editor Report · Acceptance letter]

9 Sep 2022

PONE-D-22-00071R2 

Migration of Alpine Slavs and machine learning: space-time pattern mining of an archaeological data set 

Dear Dr. Štular:

I'm pleased to inform you that your manuscript has been deemed suitable for publication in PLOS ONE. Congratulations! Your manuscript is now with our production department. 

Kind regards, 

on behalf of

Dr. Søren Wichmann 

Academic Editor

PLOS ONE